# TEACHING LLMS TO TEACH THEMSELVES BETTER INSTRUCTIONS VIA REINFORCEMENT LEARNING

## ABSTRACT

The development of Large Language Models (LLMs) often confronts challenges stemming from the heavy reliance on human annotators in the reinforcement learning with human feedback (RLHF) framework, or the frequent and costly external queries tied to the self-instruct paradigm. In this work, we pivot to Reinforcement Learning (RL)—but with a twist. Diverging from the typical RLHF, which refines LLMs following instruction data training, we use RL to directly generate the foundational instruction dataset that alone suffices for fine-tuning. Our method uses a suite of textual operations and rules, prioritizing the diversification of training datasets. It facilitates the generation of rich instructions without excessive reliance on external advanced models, paving the way for a single fine-tuning step and negating the need for subsequent RLHF stages. Our findings highlight key advantages of our approach: reduced need for human involvement and fewer model queries (**only** 5.73% **of WizardLM's total**), along with enhanced capabilities of LLMs in crafting and comprehending complex instructions compared to strong baselines, and substantially improved model privacy protection.

## 1 INTRODUCTION

In the dynamic realm of Large Language Models (LLMs), there has been a pronounced migration of their capabilities into diverse sectors, from chat robots (OpenAI, 2023; Zhao et al., 2023a; Touvron et al., 2023a) and robotics (Ahn et al., 2022; Ren et al., 2023), to autonomous driving (Fu et al., 2023; Tang et al., 2023). Amidst this broad applicability, the capacity to train with targeted instructions and pertinent responses has been integral for optimizing performance. LLMs, such as GPT-3 (Brown et al., 2020), GPT-4 (OpenAI, 2023), Llama-1 (Touvron et al., 2023a), and Llama-2 (Touvron et al., 2023b), are exemplars of this trend, showcasing enhanced capabilities when furnished with explicit human-generated instructions. Conventionally, this entailed considerable human input in both instruction creation and response generation, leading to expansive datasets for fine-tuning (Stiennon et al., 2020; Ouyang et al., 2022).

Emerging from predominantly human-instructed models, a **crucial inquiry** emerges: Can LLMs be fine-tuned to adeptly handle complex instructions *without human feedback*? The potential gains from this direction are multifaceted. Chief among them is the substantial reduction in human involvement, leading to a direct cut in costs tied to human annotations (Askell et al., 2021). Beyond the monetary aspect, such a transition also alleviates potential biases seeded by human curators (Gallegos et al., 2023). A loftier, albeit less conspicuous aim, is to amplify the quality of instructions and boost model performance. Several methodologies have surfaced in pursuit of these merits. The self-instruct method by Wang et al. (2022) stands out as a pioneering approach, which relies on LLMs to generate instructions from some guidebook. A notable recent development is the evolutionary strategy presented by Xu et al. (2023). Here, LLMs are seeded with initial instructions, gradually evolving towards generating more intricate directives within predefined constraints. Despite their method's commendable performance relative to alternative models, it necessitates a multitude of interactions with GPT-4 or ChatGPT, potentially raising concerns regarding resource demands and instruction diversity.

In this research, we propose a novel method to improve instruction quality with the principles of Reinforcement Learning (RL) (Sutton & Barto, 2018), hence enhancing LLMs' ability to comprehend and effectively execute intricate instructions without human involvement. Our approach

follows a structured yet simple process: We first train a policy enabling LLMs to generate diverse, complex instructions. We then compile a dataset of responses produced by advanced models like ChatGPT and GPT-4 in reaction to these instructions. Finally, we fine-tune a foundation LLM with this dataset, including both instructions and responses, to strengthen its capacity to process complex tasks. Our study shows that thoughtfully framing questions (via RL) is as important as, if not more than, generating responses from advanced LLMs or external sources.

Our method has two pivotal advantages. Firstly, by diminishing the dependence on human instructors (e.g., annotators and evaluators), we present a cost-effective training paradigm that supports the continuous development of capable and affordable LLMs. Secondly, our approach moderates the need for constant queries to external models. This not only translates to monetary savings but also mitigates bandwidth limitations and addresses environmental concerns with already power-hungry data centers (Dhar, 2020; Wu et al., 2022). Through comprehensive experiments, we demonstrate that our enriched training data leads to improved performance against strong baselines, simultaneously curtailing the dependency on repetitive queries to advanced models.

## 2 RELATED WORK

A multitude of studies have explored the training of language models using instructions paired with their respective responses. Notable works in this realm include GPT-3 (Brown et al., 2020), GPT-4 (OpenAI, 2023), Flan collection (Longpre et al., 2023), Flan models (Wei et al., 2021), and Alpaca (Taori et al., 2023). The prevalent methodology often requires human annotators to craft instructions and curate corresponding responses, leading to the assembly of detailed instruction-response datasets. Such datasets, exemplified by the one used in GPT-3's training (Brown et al., 2020), tend to resonate well with human preferences and markedly improve language model proficiency. However, (crowd) sourcing these datasets through human means can be not only costly but also prone to issues of bias (Gallegos et al., 2023).

Conversely, the self-instruct approach (Wang et al., 2022) charts a different course by tapping into the potential of advanced language models to autonomously generate both instructions and their responses, facilitated by the provision of predefined seeds. While this strategy alleviate the dependence on human effort, it might not consistently capture the breadth and depth of diverse instructions and responses typically achieved with human annotators. Building on the pioneering self-instruct methodology, Xu et al. (2023) introduced WizardLM, an evolutionary instruction approach. In WizardLM, initial instructions drawn from the "Evol Alpaca" dataset undergo adaptation through the amalgamation of command instructions and advanced LLMs such as GPT-4 or ChatGPT. Owing to its commendable performance in both formulating and adhering to intricate instructions, WizardLM has garnered considerable attention. However, its reliance on *random sampling* of command instructions might circumscribe the breadth and richness of instructions fed to the LLMs. Furthermore, the persistent dependence on external models—commonplace in prevailing self-instruct methods—poses concerns not just economically, but also in terms of environmental impact.

Departing from these precedents, our methodology uniquely uses RL—to generate foundational fine-tuning instruction data rather than for post-tuning refinements (such as RLHF (Stiennon et al., 2020; Ouyang et al., 2022)). We formulate a Markov Decision Process (MDP) to learn a policy for contextualized instruction manipulations that maximize instruction set diversity. While WizardLM and Tree-Instruct Zhao et al. (2023b) treat actions as ordinal choices (limiting comparability of actions and requiring optimal tree structures), we encode the continuous action space to differentiate nuances between similar actions (inherently via Q values). To solve this MDP, we use TRPO Schulman et al. (2015) (though other common methods are applicable), mitigating combinatorial complexity from sequential instruction actions and enabling iterative policy improvement (monotonically with TRPO). Remarkably, our technique reduces the query count by over 94% versus WizardLM to achieve comparable performance.

## 3 METHOD

We first train an instruction generation policy (Sec. 3.1) based on a continuous action space encoding (Sec. 3.1.1) and diversity rules as a reward function (Sec. 3.1.2). This enables teaching LLMs to

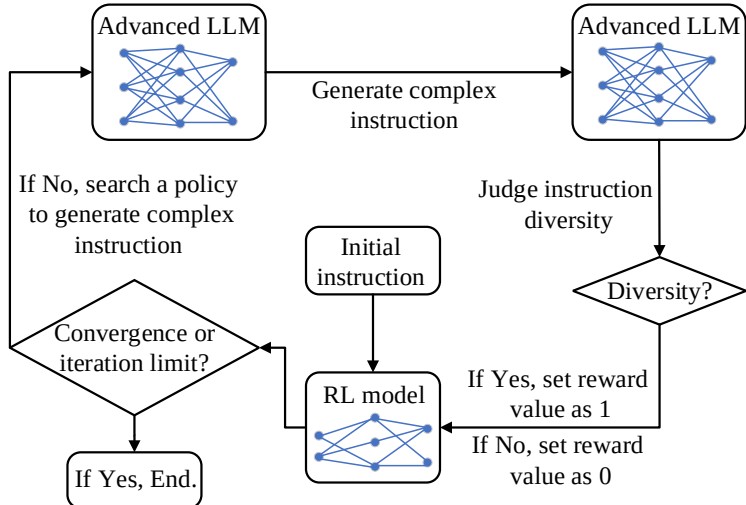

Figure 1: Train a policy with advanced LLMs.

generate complex instructions. Subsequently, using this policy and an advanced LLM, we create an instruction-response dataset to fine-tune a foundation LLM. Importantly, our enriched dataset allows skipping the typical RLHF phase, equipping LLMs to adeptly generate and comprehend complex instructions with instruction fine-tuning alone. Also, the instruction policy (obtained in Sec. 3.1) is transferable for aligning various foundation LLMs such as llama-1-chat-7b and llama-2-chat-7b (see Secs. 4.2.2 and 4.2.1).

## 3.1 TRAIN A POLICY

As depicted in Figure 1, we employ the open-source WizardLM-13b model[1,2] as a proxy environment to train our policy. As a cost-effective model, WizardLM-13b can capture instruction nuances to provide reward signals (Sec. 3.1.2), rather than serving knowledge distillation needs. Furthermore, we leverage the Alpaca dataset[3] as the primary source of initial instructions, contributing to the comprehensive training process.

The training procedure comprises several key stages. First, we select a single initial instruction from the Alpaca dataset, such as "Give three tips for staying healthy." This chosen instruction is then input into our RL model. Second, we evaluate the RL model's performance by assessing the convergence of reward signals. The policy training concludes if the rewards demonstrate convergence or the iteration limit is reached. Following this iterative training phase, we enlist an advanced language model to generate complex instructions using a tailored action space (Sec. 3.1.1). In the final phase, the same advanced LLM assesses the diversity of the generated instructions to ensure richness and variety (Sec. 3.1.2).

### 3.1.1 ACTION SET

Inspired by WizardLM (Xu et al., 2023), we define several actions to generate complex instructions: *"breadth action", "add constraint", "deepening", "concretizing", "increase reasoning steps", "complicate input"*. While the actions resemble WizardLM, we map the discrete set into a continuous output space for an instruction generation policy. This enables inherently capturing contextual nuances for direct comparison between actions via their Q values. The details of each action are in Appendix A.1.

### 3.1.2 REWARD SETTINGS

In this section, consider a judicial prompt $g$. If $g$ is characterized as "equal", the corresponding reward $r$ is designated as 0. Otherwise, $r$ is set to 1. It is noteworthy that the judgment is determined

---

[1]`https://github.com/nlpxucan/WizardLM`

[2]`https://huggingface.co/WizardLM/WizardLM-13B-V1.2`

[3]`https://huggingface.co/datasets/tatsu-lab/alpaca`

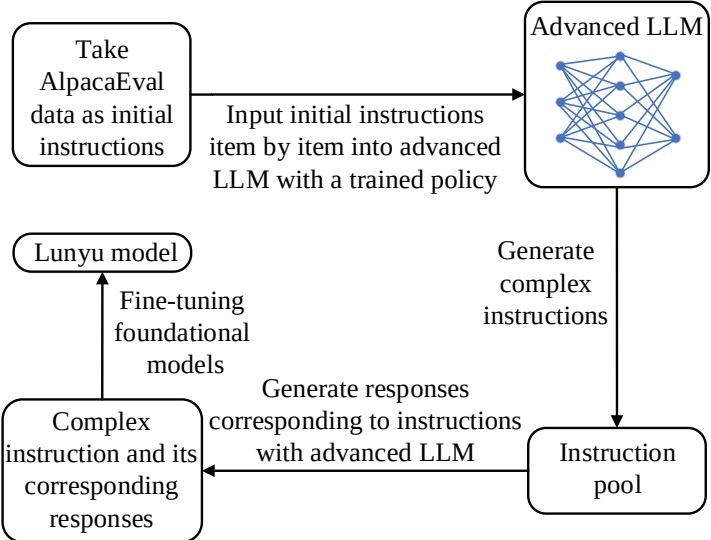

Figure 2: Fine-tune a foundational model as the Lunyu model.

by advanced models such as WizardLM-13b. For instance, when there is congruence between the initial and subsequent instructions, the reward $r$ is quantified as 0. In contrast, a lack of congruence yields a reward value of 1.

*judicial prompt = """ Here are two Instructions to ChatGPT AI, do you think they are equal to each other, which meet the following requirements: 1. They have same constraints and requirments. 2. They have same depth and breadth of the inquiry. The First Instruction: """ +instruction + """". The Second Instruction: """ + state + """". Your Judgement (Must Just only answer: Equal or Not Equal. No need to explain the reason; 'Equal' and 'Not Equal' are not allowed to appear simultaneously.): """.*

## 3.2 FINE-TUNE A FOUNDATIONAL MODEL

As delineated in Figure 2, a judicious approach is adopted to optimize cost-efficiency considerations during the phase dedicated to fine-tuning foundational models. Specifically, we limit the initial instructions to the Alpaca dataset.

The fine-tuning process is executed as follows: We first input the selected instructions into an advanced model like ChatGPT or GPT-4. Next, our trained instruction generation policy produces complex instructions by expanding on the initial ones, which are added to the instruction pool. Concurrently, the advanced model produces responses to these complex instructions. With the instructions and responses in hand, we fine-tune foundation models like llama-1-chat-7b and llama-2-chat-7b via supervised learning—a prudent, affordable way to enhance their capabilities.

---

**Algorithm 1** Lunyu: Enhencing LLMs to Follow Complex Instructions through RL.

---
1: Design a set $A$ of actions for policy search.
2: Map the actions into a discrete value-based action space $S$.
3: Leverage TRPO (Schulman et al., 2015) and an advanced model (here we use WizardLM-13b) to search for a policy that can help generate diverse instructions.
4: Utilize the trained policy to teach advanced LLMs such as GPT-4 or ChatGPT to generate complex instructions and corresponding responses with initial instructions.
5: Fine-tune a foundational language model with the generated instructions and corresponding responses.

---

## 3.3 A PRACTICAL ALGORITHM

As mentioned, our process culminates in the pragmatic Algorithm 1. Building on this, we fine-tune a novel LLM called "Lunyu"—honoring Confucius' Analects by generating quality instructions reflecting ideals of iterative learning and proper teaching. Within this, we have flexibility to choose

established RL methods for policy training, specifically Trust Region Policy Optimization (TRPO) (Schulman et al., 2015), though it can have longer training times versus alternatives like Proximal Policy Optimization (PPO). We select TRPO for its rigorous advantage function handling and theoretical guarantees of monotonic improvement. Importantly, our algorithm judiciously leverages advanced LLMs only in two key steps: WizardLM-13B solely serves as the proxy environment for policy search (Step 3), while ChatGPT/GPT-4 sparingly generate instruction responses (Step 4).

## 4 EXPERIMENTS

### 4.1 TRAIN A POLICY TO GENERATE COMPLEX INSTRUCTIONS

We deploy a policy that is designed to orchestrate a trajectory consisting of six distinct instruction actions, in accordance with the principles delineated in Algorithm 1 to facilitate the generation of intricate instructions by LLMs. A noteworthy aspect of our investigation involves a comparative assessment of the data quality resulting from the utilization of our policy, in contrast to the approach adopted by WizardLM (Xu et al., 2023), which relies upon random sampling for data generation.

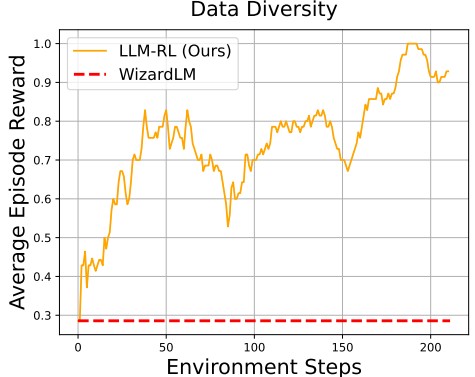

Figure 3: Compare our method with WizardLM in terms of data diversity.

It bears emphasizing that our instruction actions are randomly initialized, but as training progresses, our policy iteratively learns to enable advanced LLMs to produce increasingly complex and diverse instructions. Our findings in Figure 3 demonstrate the superior data quality achieved. Also, in designing instruction actions, the "breadth action" is a single action regarding breath thoughts. Thus, we insert the action into the middle of our trajectory to enhance breadth instructions after training a policy.

The main computation overhead is learning the instruction policy on the relatively small WizardLM-13b in less than 1 hour (with 371 total queries). This results in a transferable policy that reduces alignment cost across models compared to RLHF's per-model RL. Offloading to policy learning provides an instruction set for joint tuning and alignment—a substantial benefit over tuning-only data usage.

### 4.2 COMPARISON EXPERIMENTS ON LM EVAL BENCHMAKRS

To comprehensively examine the effectiveness of our method, we carry out experiments on LM-Eval benchmark[4,5], the LM-Eval benchmark stands as a prominent tool for assessing LLM performance. It encompasses a suite of sub-benchmarks: (1) AI2 Reasoning Challenge (ARC) benchmark (Clark et al., 2018): The benchmark introduces a fresh question set, text corpus, and baselines, all strategically curated to foster and propel AI research in the realm of advanced question answering, setting a significantly higher bar for knowledge and reasoning capabilities compared to previous challenges. (2) HellaSwag benchmark (Zellers et al., 2019): The benchmark introduces a challenging dataset, revealing that even state-of-the-art models struggle with commonsense inference, as evidenced by the significant performance gap between humans ($95\%$ accuracy) and models ($48\%$), achieved through adversarial filtering, a robust data collection paradigm that selects adversarial machine-generated wrong answers by scaling up the length and complexity of dataset examples to a 'Goldilocks' zone where the text generated is absurd to humans yet often misclassified by models. (3) Massive Multi-task Language Understanding (MMLU) benchmark (Hendrycks et al., 2020): The benchmark serves as a comprehensive assessment of a text model's multitask accuracy, encompassing a total of 57 distinct tasks. These tasks span various domains, including elementary mathematics, US history, computer science, law, and others. Achieving a high level of accuracy on this benchmark necessitates

---

[4]https://huggingface.co/spaces/HuggingFaceH4/open_llm_leaderboard
[5]https://github.com/EleutherAI/lm-evaluation-harness

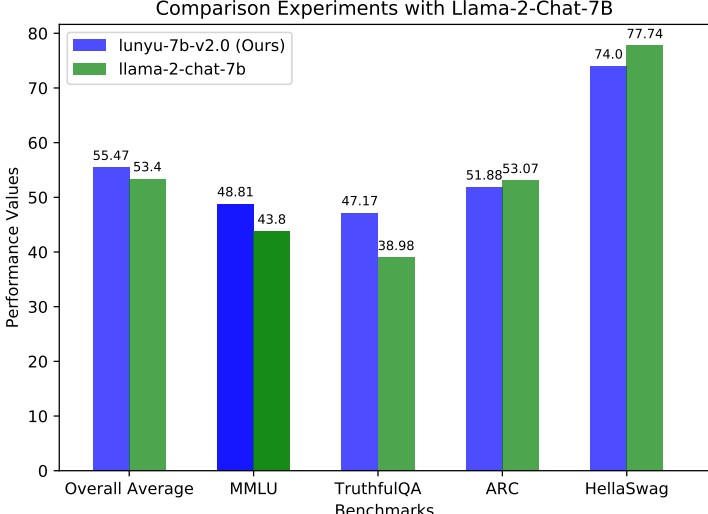

Figure 4: Compare our method with llama-2-chat-7b on LM-Eval Benchmarks.

a strong grasp of world knowledge and strong problem-solving capabilities. (4) Truth Question-Answering (TruthQA) benchmark (Lin et al., 2022): The benchmark encompasses a diverse array of 817 questions distributed across 38 distinct categories, encompassing a wide spectrum of domains such as health, law, finance, and politics.

In our comparison experiments, we take the same setting as the LM-Eval benchmarks shown, e.g., 25 shots for ARC, 10 shots for HellaSwag, 5 shots for MMLU, and 0 shot for TruchfulQA.

### 4.2.1 COMPARE WITH A LAMA-2-CHAT-7B MODEL

We conducted a comparative analysis in the evaluation involving our methodology and Llama-2-chat-7b[6]. As illustrated in Figure 4, we trained a model, denoted as "lunyu-7b-v2.0," based on the llama-2-chat-7b architecture using our dataset, encompassing a total of 15,392 instructions along with their corresponding responses. The results of our experimental evaluation reveal noteworthy insights: our model outperforms the llama-2-chat-7b model across several performance metrics, demonstrating superior overall performance. Notably, our methodology exhibits a particularly substantial advantage in tasks related to MMLU and TruthQA, where it exhibits a significant performance edge over the llama-2-chat-7b model.

### 4.2.2 COMPARE WITH WIZARDLM-7B AND LLAMA-1-7B MODELS

Furthermore, we extended our comparative analysis to a llama-1-7b model denoted "lunyu-7b-v1.1," trained on our dataset of 17,878 instruction-response pairs. As shown in Figure 5, our method shows superior average performance over llama-1-7b models and on par with wizard-7b models [7,8]. We also fine-tuned a "lunyu-7b-v1.0" model on our previous 15,392-sample dataset, attaining performance comparable to wizard-7b. The slightly lower HellaSwag/ARC results seem to originate from the initial Alpaca instruction set prioritizing multi-tasking over specialization, evidenced by similar WizardLM outcomes (Figs. 4 and 5). See AppendixA.3 for details.

It is crucial to underscore a salient aspect of our methodology in relation to data utilization. The dataset employed for training our model is approximately one-fourteenth the size of the dataset utilized by wizardlm, as illustrated in Figure 6 (a). Furthermore, Figure 6 (b) highlights the discernible difference in the query count posed to GPT models between our method and wizardlm, with the

---

[6]https://huggingface.co/meta-llama/Llama-2-7b-chat-hf

[7]https://huggingface.co/TheBloke/wizardLM-7B-HF

[8]In our experimental setup, all models are configured with a float16 format. We compare to wizard-7b since both this approach and ours use llama-1-7b as the base model. Notably, WizardLM7B queried ChatGPT 624,000 times for responses, whereas our method queried open-source WizardLM13B 371 times during policy training and ChatGPT 35,756 times. As WizardLM13B has similar capabilities to ChatGPT, our total queries are substantially fewer. Therefore, we believe the comparison is fair in terms of matched base model and vastly lower query amount.

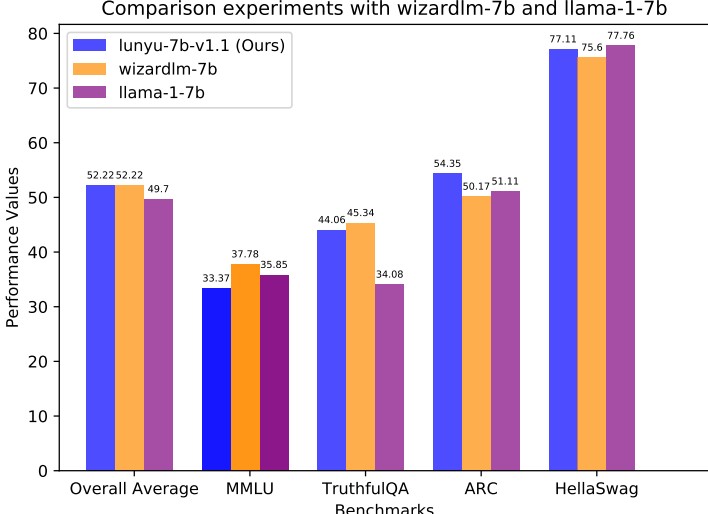

Figure 5: Compare our method with wizardlm 7B and llama-1-7b on LM-Eval Benchmarks.

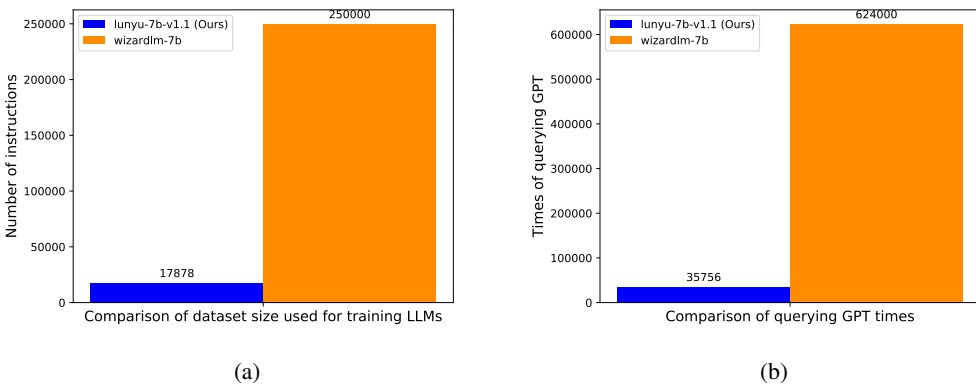

Figure 6: Compare with wizardlm 7B on dataset size used for training LLMs and querying times of advanced LLMs.

latter soliciting GPT models for responses at a rate seventeen times higher. This marked contrast underscores the cost-effectiveness of our data collection approach, which mitigates the expenses associated with dataset acquisition. It highlights that our method is a more economically viable and sustainable strategy for training LLMs.

### 4.3 EXPERIMENTS OF MODEL PRIVACY ATTACK

Our objective is to improve model performance while enhancing data privacy, even with limited data. Notably, the Lunyu methodology can effectively mitigate privacy leakage risks typically associated with large datasets. Our experiments clearly demonstrate substantially enhanced privacy protection over the baseline model. See Appendix A.2 for details.

### 4.4 GENERATE INSTRUCTIONS WITH A TRAINED POLICY

Capitalizing on the policy we have trained, complex instructions are generated by synergizing initial instructions with GPT-4 or GPT-3.5. The related cases are in subsections 4.4.1 and 4.4.2. It is noteworthy that these initial instructions are culled from a well-established and widely-recognized dataset, namely the Alpaca dataset[9].

### 4.4.1 GENERATE INSTRUCTIONS WITH GPT-4 AND OUR POLICY

> An initial instruction:"Build a bridge between two islands."

---

[9]https://github.com/tatsu-lab/alpaca_eval.git

Generated instructions: (1) "Construct a bridge connecting two islands while ensuring it can withstand the environmental conditions and weight capacity requirements of the region." (2) "Devise an interconnected subterranean network connecting two archipelagos, factoring in geological characteristics, pressure endurance, requisite airflow for secure passage, and minimal ecological disruption." (3) "Devise a schematic representation of an underwater tunnel system linking numerous island chains, factoring in the topography of the ocean floor, resilience to hydrostatic forces, adequate ventilation for commuters, minimized environmental disruptions, and cost-effective building methodologies."

An initial instruction: "Describe the economic system of the United States."

Generated instructions: (1) "Elucidate the economic framework prevalent in the United States, taking into account its relationship with environmental sustainability." (2) "Examine the monetary structure dominating in the European Union, while considering its connection to ecological preservation." (3) "Investigate the prevailing monetary framework within the European Union, taking into account its association with environmental conservation, and evaluate the impact of one additional factor, such as economic growth or political stability."

For more cases, please see Appendix A.4.

### 4.4.2 GENERATE INSTRUCTIONS WITH GPT-3.5 AND OUR POLICY

An initial instruction: "Find a unique way to use a rubber band."

Generated instructions: (1) "Discover an unconventional and singular application for an elastic band, ensuring that it adheres to environmentally friendly practices." (2) "Develop a groundbreaking and ecologically considerate utilization for a stretchable loop, ensuring adherence to the principles of enduring viability, while elaborating on the prospective enduring environmental advantages it may offer." (3) "Devise a groundbreaking, environmentally-conscious application for a scalable band, contemplating prolonged durability and sustainability, while elaborating on the enduring ecological advantages it could potentially offer." For more cases, please see Appendix A.5.

### 4.5 LUNYU-7B-V2.0 PERFORMANCE ON RANDOMLY INSTRUCTIONS

In this evaluation, we present a scenario where a random instruction is provided, and the subsequent response is generated by our model, referred to as lunyu-7b-v2.0. Notably, the model performs exceedingly well in providing a well-structured and accurate response to the given instruction. More great cases are provided in Appendix A.6.

Instruction: "Write a simple guide for uploading the bibliography database on Overleaf."

Lunyu-7b-v2.0 output: "Sure, I'd be happy to help you with that! Here's a step-by-step guide on how to upload a bibliography database on Overleaf: 1. First, you'll need to create an account on Overleaf if you don't already have one. Once you're logged in, click on the "New Project" button to start a new project. 2. Next, you'll need to create a new bibliography database. To do this, click on the "Bibliography" tab in the top menu bar and then click on "Add New Bibliography". 3. In the "Add New Bibliography" window, you'll need to enter the details of your bibliography database. This includes the title, author, publication date, and any other relevant information. 4. Once you've entered all the details, you can click on the "Save" button to save your bibliography database. 5. Next, you'll need to upload your bibliography database to Overleaf. To do this, click on the "Upload" button in the top menu bar and then select the file you want to upload. 6. Once you've uploaded your file, you can click on the "Save" button to save your bibliography database. 7. Finally, you can use your bibliography database in your Overleaf project by clicking on the "Bibliography" tab in the top menu bar and then selecting the bibliography database you want to use.
That's it! With these steps, you should be able to upload your bibliography database on Overleaf. If you have any questions or need further assistance, feel free to ask."

### 4.6 COMPARISON EXPERIMENTS OF SOLVING A MATH PROBLEM

We conduct comparison experiments with several strong baseline models on mathematical tasks. The experimental procedure entails inputting instructions into LLMs, with the LLMs subsequently generating responses aligned with the provided instructions. More results are introduced in Appendix A.7.

Instruction: "Given that $f(x) = 5x^3 - 2x + 3$, find the value of f(2)."

Lunyu-7b-v2.0 output: "Hello! I'm here to help you with your question. To find the value of f(2), we need to substitute x = 2 into the function $f(x) = 5x^3 - 2x + 3$.
1. $5(2)^3 - 2(2) + 3 = 5(8) - 2 + 3 = 40 - 2 + 3 = 39$
So, the value of f(2) is 39."
The output is correct.

Llama-2-chat-13b output: "Hello! I'd be happy to help you with that! To find the value of f(2), we need to plug 2 into the function f(x) = $5x^3$ - 2x + 3.
First, let's evaluate the expression inside the parentheses:
$5x^3$ - 2x + 3
When x = 2, we have: $5(2)^3$ - 2(2) + 3
= 5(8) - 2(2) + 3
= 40 - 4 + 3
= 36
So, f(2) = 36.
Is there anything else you'd like to know?"
The output is wrong.

ChatGPT output: "To find the value of f(2), we simply need to substitute 2 in place of x in the given function and simplify.
f(2) = $5(2)^3$ - 2(2) + 3
f(2) = 5(8) - 4 + 3
f(2) = 40 - 1
f(2) = 39
Therefore, the value of f(2) is 39."
The output is correct.

## 5 CONCLUSION

In this study, we depart from the traditional RLHF paradigm and introduce a method that amplifies instruction quality while significantly cutting the costs linked to querying proprietary LLMs such as ChatGPT or GPT-4, a feat realized through strategic RL application in autonomous instruction set generation. Leveraging this methodology, we refine a foundational model and conduct comprehensive experiments to assess its efficacy. Remarkably, the LLM trained under this framework rivals the performance of the acclaimed WizardLM, despite being constrained by a significantly smaller dataset and fewer query instances: our dataset amounts to a mere $6.75\%$ of WizardLM's, and the query counts to ChatGPT or GPT-4 are only $5.73\%$ of what WizardLM uses. This highlights the economic and sustainable advantages of our approach, underscoring its potential to enhance data quality within budgetary confines.

Beyond mere practical implications, our findings question the conventional two-stage LLM training pipeline, suggesting that perhaps it's possible to train proficient LLMs without the necessity of human feedback. It beckons a rethinking of the pivotal role humans play in LLM training, urging a more judicious deployment of human resources to truly critical facets of the training process.

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

CONTENT

# A  APPENDIX

## A.1  ACTION SET

breadth action:

evol prompt = """"I want you act as a Prompt Creator. Your goal is to draw inspiration from the #Given Prompt# to create a brand new prompt. This new prompt should belong to the same domain as the #Given Prompt# but be even more rare. The LENGTH and difficulty level of the #Created Prompt# should be similar to that of the #Given Prompt#. Don't repeat the conditions and requirements in the response, and Don't disclose your role. The Prompt Rewriter Must not give the introduction and explain the reason, the Prompt Rewriter must just give the most relevant response. This new prompt should not exceed 2048 words. The #Created Prompt# must be reasonable and must be understood and responded by humans. '#Given Prompt#', '#Created Prompt#', 'given prompt' and 'created prompt' are not allowed to appear in #Created Prompt#. #Given Prompt#: """ + instruction

add constraints:

evol prompt = """" I want you act as a Prompt Rewriter. Your objective is to rewrite a given prompt into a more complex version to make those famous AI systems (e.g., chatgpt and GPT4) a bit harder to handle. But the rewritten prompt must be reasonable and must be understood and responded by humans. Your rewriting cannot omit the non-text parts such as the table and code in #Given Prompt#:. Also, please do not omit the input in #Given Prompt#. Don't repeat the conditions and requirements in the response, and Don't disclose your role. The Prompt Rewriter Must not give the introduction and explain the reason, the Prompt Rewriter must just give the most relevant response. This new prompt should not exceed 2048 words. You SHOULD complicate the given prompt using the following method: Please add one more constraints/requirements into #Given Prompt# You should try your best not to make the #Rewritten Prompt# become verbose, #Rewritten Prompt# can only add or replace 10 to 20 words into #Given Prompt#. '#Given Prompt#', '#Rewritten Prompt#', 'given prompt' and 'rewritten prompt' are not allowed to appear in #Rewritten Prompt#. #Given Prompt#: """ + instruction

deepening:

evol prompt = """" I want you act as a Prompt Rewriter. Your objective is to rewrite a given prompt into a more complex version to make those famous AI systems (e.g., chatgpt and GPT4) a bit harder to handle. But the rewritten prompt must be reasonable and must be understood and responded by humans. Your rewriting cannot omit the non-text parts such as the table and code in #Given Prompt#:. Also, please do not omit the input in #Given Prompt#. Don't repeat the conditions and requirements in the response, and Don't disclose your role. The Prompt Rewriter Must not give the introduction and explain the reason, the Prompt Rewriter must just give the most relevant response. This new prompt should not exceed 2048 words. You SHOULD complicate the given prompt using the following method: If #Given Prompt# contains inquiries about certain issues, the depth and breadth of the inquiry can be increased. You should try your best not to make the #Rewritten Prompt# become verbose, #Rewritten Prompt# can only add 10 to 20 words into #Given Prompt#. '#Given Prompt#', '#Rewritten Prompt#', 'given prompt' and 'rewritten prompt' are not allowed to appear in #Rewritten Prompt#. #Given Prompt#: """ + instruction



**concretizing**:

evol prompt = """ I want you act as a Prompt Rewriter. Your objective is to rewrite a given prompt into a more complex version to make those famous AI systems (e.g., chatgpt and GPT4) a bit harder to handle. But the rewritten prompt must be reasonable and must be understood and responded by humans. Your rewriting cannot omit the non-text parts such as the table and code in #Given Prompt#:. Also, please do not omit the input in #Given Prompt#. Don't repeat the conditions and requirements in the response, and Don't disclose your role. The Prompt Rewriter Must not give the introduction and explain the reason, the Prompt Rewriter must just give the most relevant response. This new prompt should not exceed 2048 words. You SHOULD complicate the given prompt using the following method: Please replace general concepts with more specific concepts. You should try your best not to make the #Rewritten Prompt# become verbose, #Rewritten Prompt# can only add 10 to 20 words into #Given Prompt#. '#Given Prompt#', '#Rewritten Prompt#', 'given prompt' and 'rewritten prompt' are not allowed to appear in #Rewritten Prompt#. #Given Prompt#: """ + instruction





**increase reasoning steps**:

evol prompt = """ I want you act as a Prompt Rewriter. Your objective is to rewrite a given prompt into a more complex version to make those famous AI systems (e.g., chatgpt and GPT4) a bit harder to handle. But the rewritten prompt must be reasonable and must be understood and responded by humans. Your rewriting cannot omit the non-text parts such as the table and code in #Given Prompt#:. Also, please do not omit the input in #Given Prompt#. Don't repeat the conditions and requirements in the response, and Don't disclose your role. The Prompt Rewriter Must not give the introduction and explain the reason, the Prompt Rewriter must just give the most relevant response. This new prompt should not exceed 2048 words. You SHOULD complicate the given prompt using the following method: If #Given Prompt# can be solved with just a few simple thinking processes, you can rewrite it to explicitly request multiple-step reasoning. You should try your best not to make the #Rewritten Prompt# become verbose, #Rewritten Prompt# can only add 10 to 20 words into #Given Prompt#. '#Given Prompt#', '#Rewritten Prompt#', 'given prompt' and 'rewritten prompt' are not allowed to appear in #Rewritten Prompt#. #Given Prompt#: """ + instruction





**complicate input**:

evol prompt = """ I want you act as a Prompt Rewriter. Your objective is to rewrite a given prompt into a more complex version using dataformat to make those famous AI systems (e.g., chatgpt and GPT4) more difficult to handle. But the rewritten prompt must be reasonable and must be understood and responded by humans. Don't repeat the conditions and requirements in the response, and Don't disclose your role. The Prompt Rewriter Must not give the introduction and explain the reason, the Prompt Rewriter must just give the most relevant response. This new prompt should not exceed 2048 words. The Given Prompt: """+ instruction



## A.2 EXPERIMENTS OF MODEL PRIVACY ATTACK

In AI safety, it has increasingly adopted data synthesizers designed to produce differentially private datasets to mitigate the risk of inadvertent data leakage (Dong et al., 2022). These datasets serve as the foundational element for training machine learning algorithms. However, it presents a dilemma wherein practitioners have to choose between large training data and data privacy. Addressing this problem, we introduce the "Lunyu" methodology, a novel approach that tries to handle the dilemma of large training data and data privacy. It aims to enhance the performance of training models while ensuring improved data privacy, even with limited data. Notably, the Lunyu methodology mitigates the risk of data privacy leakage often associated with using large datasets.

As illustrated in Figure 7, we have conducted a series of membership inference attack (Shokri et al., 2017; Carlini et al., 2022) experiments to assess our model's privacy performance rigorously. Our model exhibits a Receiver Operating Characteristic (ROC) curve that closely approximates random

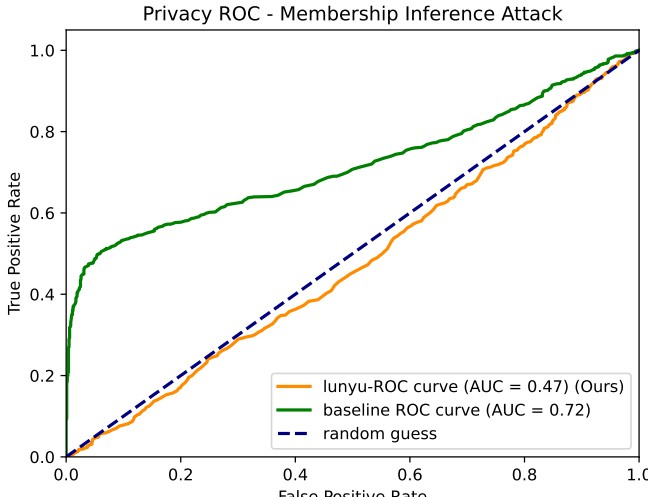

Figure 7: Privacy Attacks on the Model: Our model demonstrates strong privacy protection performance. The more closely the ROC curve of the model aligns with random guessing, and the closer the AUC value of the model approaches 0.5, the stronger the indication of improved privacy protection by the model.

guessing, yielding an Area Under the Curve (AUC) value of 0.47 in this evaluation. Conversely, the baseline model, trained on a dataset comprising 44,000 samples, displays a notable deviation from random guessing, with an AUC value of 0.72.

In the context of model privacy assessment, a closer proximity of the ROC curve to the random guess curve indicates better model privacy protection performance, while an AUC value approaching 0.5 further suggests better model privacy protection (Ye et al., 2022). The experiment results demonstrate the substantial enhancement in model privacy protection performance achieved by our model relative to the baseline model.

### A.3 COMPARISON EXPERIMENTS WITH LUNYU-7B-V1.0

As shown in Figure 8, we employed a previously curated dataset comprised of 15,392 paired instructions and responses to refine the performance of a llama-1-7b model. This refined model, designated as "lunyu-7b-v1.0," has been meticulously evaluated and benchmarked against the performance metrics characteristic of the wizard-7b models. The comprehensive assessment entailed a rigorous analysis of various parameters and performance indices to ensure an unbiased and accurate comparison.

Our findings elucidate that "lunyu-7b-v1.0" manifests a performance equilibrium, demonstrating metrics that are comparable with those exhibited by the wizard-7b models. This almost parity in performance, achieved amidst the context of a reduced dataset, accentuates the efficacy of our methodology and posits "lunyu-7b-v1.0" as a competitive alternative in the expansive landscape of language models.

As depicted in Figures 9 (a) and (b), the utilization of a concise yet diverse dataset comprising 15,392 instructional pairs substantiates the possibility of achieving optimal performance without the exigency of voluminous data. This revelation not only affirms the efficiency and effectiveness of our approach but also underscores a significant stride towards more sustainable and economical practices in developing and deploying advanced language models.

### A.4 GENERATE INSTRUCTIONS WITH GPT 4

An initial instruction:
"Build a bridge between two islands."

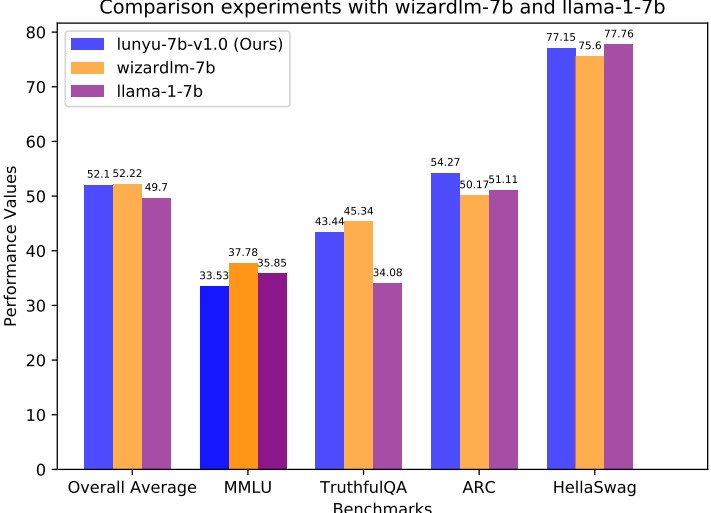

Figure 8: Compare our method with wizardlm 7B and llama-1-7b on LM-Eval Benchmarks.

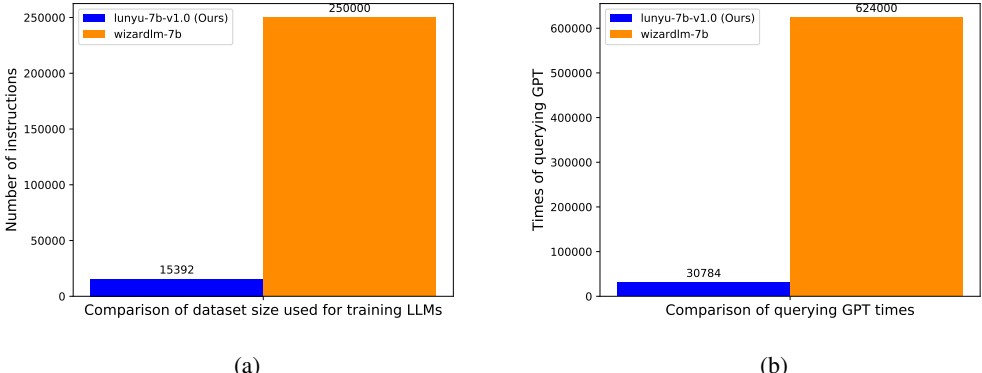

Figure 9: Compare with wizardlm 7B on querying times of advanced LLM models.

Evolved instructions:
(1) "Construct a bridge connecting two islands while ensuring it can withstand the environmental conditions and weight capacity requirements of the region.",

(2) "Design a tunnel system that links two archipelagos, taking into consideration the geological features, pressure resistance, and necessary ventilation for safe travel.",

(3) "Devise an interconnected subterranean network connecting two archipelagos, factoring in geological characteristics, pressure endurance, requisite airflow for secure passage, and minimal ecological disruption.",

(4) "Design an interlinked underground system joining two island chains, incorporating geological features, resistance to pressure, adequate ventilation for safe traversal, and minimal environmental impact, while ensuring cost-efficiency.",

(5) "Create a blueprint for a subaquatic tunnel network connecting multiple archipelagos, taking into account underwater terrain, structural integrity against water pressure, sufficient air circulation for travelers, reduced ecological disturbance, and budget-conscious construction.",

(6) "Devise a schematic representation of an underwater tunnel system linking numerous island chains, factoring in the topography of the ocean floor, resilience to hydrostatic forces, adequate ventilation for commuters, minimized environmental disruptions, and cost-effective building methodologies."

An initial instruction:
"Describe the economic system of the United States."

Evolved instructions:
(1) "Elucidate the economic framework prevalent in the United States, taking into account its relationship with environmental sustainability.",

(2) "Examine the monetary structure dominating in the European Union, while considering its connection to ecological preservation.",

(3) "Investigate the prevailing monetary framework within the European Union, taking into account its association with environmental conservation, and evaluate the impact of one additional factor, such as economic growth or political stability."

A initial instruction:
"Convert 28 cm to inches."

Evolved instructions:
(1) "In order to convert 28 centimeters into inches, taking into account that 1 inch is equivalent to 2.54 centimeters, and provide the result with two decimal places of precision.",

(2) "Calculate the length in inches of a 35-centimeter object, considering that 1 inch equals 2.54 centimeters, and present your answer rounded to two decimal places.",

(3) "Determine the measurement in inches for an object with a length of 35 centimeters, taking into account the conversion factor of 1 inch being equivalent to 2.54 centimeters, and provide the resulting value with a precision of two decimal points.",

(4) "Ascertain the dimensions in inches of an item possessing a length of 35 centimeters, factoring in the conversion ratio of 1 inch corresponding to 2.54 centimeters, and supply the consequent value, ensuring accuracy to a couple of decimal places.",

(5) "Determine the weight in pounds of an object with a mass of 45 kilograms, considering the conversion rate of 1 kilogram equaling 2.20462 pounds, and provide the resulting value with precision up to two decimal points.",

(6) "Calculate the weight in pounds of an item possessing a mass of 45 kg, taking into account the conversion factor of 1 kg being equivalent to 2.20462 lbs, and present the outcome with accuracy to two decimal places."

A initial instruction:
"Predict the weather in Los Angeles tomorrow."

Evolved instructions:
(1) "Forecast the atmospheric conditions for the city of Los Angeles, California, specifically for the upcoming day.",

(2) "Predict the weather patterns for San Francisco, California, focusing on the next 24 hours.",

(3) "Forecast the atmospheric conditions for San Francisco, California, with a particular emphasis on the upcoming 24-hour period, taking into account temperature, precipitation, and wind patterns.",

(4) "Predict the meteorological conditions in San Francisco, California, focusing specifically on the subsequent 24-hour timeframe, incorporating temperature, precipitation, wind patterns, and humidity levels.",

(5) "Analyze the atmospheric conditions in Seattle, Washington, for the upcoming 48-hour period, concentrating on factors such as air pressure, cloud coverage, visibility, and dew point.",

(6) "Examine and provide a detailed assessment of the meteorological conditions and atmospheric parameters in Seattle, Washington, for the subsequent 48-hour time frame. This analysis should predominantly focus on elements such as barometric pressure, the extent of cloud cover, range of visibility, and the dew point temperature."

A initial instruction:
"Tell me about the seasons in the temperate forest biome."

Evolved instructions:
(1) "Elaborate on the distinct seasonal variations observed within the deciduous forest biome, encompassing climate and ecological adaptations.",

(2) "Discuss the unique cyclic changes experienced in the coniferous forest ecosystem, focusing on weather patterns and the adaptive strategies of its flora and fauna.",

(3) "Elaborate on the distinct cyclical transformations occurring within the coniferous forest ecosystem, emphasizing meteorological fluctuations and the specialized adaptation techniques employed by its plant and animal species.",

(4) "Examine the unique cyclical transformations taking place in the coniferous forest ecosystem, with an emphasis on meteorological fluctuations, the specialized adaptation techniques employed by its plant and animal species, and the interdependence between biotic and abiotic factors.",

(5) "Analyze the distinct seasonal changes occurring in the alpine tundra ecosystem, focusing on the influence of climatic variations, the specific survival strategies utilized by its flora and fauna, and the mutual relationships between living and non-living elements.",

(6) "Examine the unique seasonal transformations in the alpine tundra biome, emphasizing the impact of climatic fluctuations, the specialized adaptation mechanisms employed by its plant and animal species, and the interdependent connections between biotic and abiotic components."

A initial instruction:
"Generate a list of items for a vegetarian Thanksgiving dinner."

Evolved instructions:
(1) "Devise a strategy for a Thanksgiving feast that is not only economical but also incorporates at least one vegetarian dish to accommodate diverse dietary preferences.",

(2) "Plan a budget-friendly Easter dinner that includes a vegan main course option to cater to various dietary needs.",

(3) "Devise an economical Easter dinner plan, encompassing a vegan main course alternative and an additional gluten-free side dish, in order to accommodate a range of dietary preferences.",

(4) "Construct an affordable Easter dinner strategy, incorporating a vegan main course option and an extra gluten-free side dish, while also ensuring the inclusion of a low-sodium appetizer, to cater to a variety of dietary inclinations.",

(5) "Design a budget-friendly Easter meal plan that features a plant-based centerpiece and an additional side dish without gluten, as well as an appetizer low in salt content, to accommodate various dietary preferences.",

(6) "Devise an economical Easter meal plan, encompassing a plant-based focal dish, an accompanying side free of gluten, and an appetizer with minimal sodium, while also incorporating a dessert that is devoid of nuts, catering to a wide array of dietary inclinations."

A initial instruction:
"Name two characters in the novel Moby Dick."

Evolved instructions:
(1) "Identify a pair of individuals featured in Herman Melville's literary work, Moby Dick, while also specifying the roles they play within the narrative.",

(2) "Examine two characters from Nathaniel Hawthorne's novel, The Scarlet Letter, and describe the significance of their roles in the story.",

(3) "Delve into the intricacies of two characters from Nathaniel Hawthorne's literary work, The Scarlet Letter, and elucidate the importance of their contributions to the overarching narrative and themes within the novel.",

(4) "Examine and dissect the complexities of a pair of characters featured in Nathaniel Hawthorne's eminent piece of literature, The Scarlet Letter. Provide a comprehensive analysis of the significance of their individual and collective roles in augmenting the overarching storyline and the prevailing motifs embedded throughout the novel.",

(5) "Analyze and explore the intricate dynamics between two characters from F. Scott Fitzgerald's renowned work, The Great Gatsby. Offer a thorough evaluation of how their individual and combined contributions enhance the central narrative and reinforce the dominant themes woven throughout the story.",

(6) "Delve into the complex interplay between two characters from F. Scott Fitzgerald's acclaimed novel, The Great Gatsby, and provide an in-depth assessment of how their singular and mutual interactions augment the core storyline while bolstering the predominant motifs interspersed within the tale."

A initial instruction:
"Name 6 components of an artificial neural network".

Evolved instructions:

(1) "Identify 6 distinct elements that constitute an artificial neural network, including various layers and associated features.",

(2) "Enumerate half a dozen unique components that form the structure of an artificial neural network, encompassing diverse stratums and their corresponding characteristics.",

(3) "Enumerate six distinct elements constituting the architecture of an artificial neural network, encompassing various hierarchical layers and their corresponding attributes, such as input, hidden, and output layers, along with activation functions, weights, and biases.",

(4) "List six unique components that make up the structure of a deep learning model, including different levels of organization and their associated features, such as convolutional, recurrent, and fully connected layers, as well as loss functions, optimization algorithms, and regularization techniques.",

(5) "Enumerate six distinct elements constituting the architecture of a deep learning paradigm, encompassing various organizational strata and their pertinent characteristics, like convolutional, recurrent, and fully connected strata, in addition to loss computation methods, optimization procedures, and regularization approaches.",

(6) "Identify seven unique components that form the structure of an advanced machine learning framework, including different layers and their relevant features, such as attention, transformer, and radial basis function layers, along with performance evaluation techniques, training strategies, and hyperparameter tuning methods."

## A.5 Generate Instructions with GPT 3.5

An initial instruction:
"Find a unique way to use a rubber band."

Evolved instructions:

(1) "Discover an unconventional and singular application for an elastic band, ensuring that it adheres to environmentally friendly practices.",

(2) "Invent a unique and eco-conscious use for a rubber band, while ensuring that its implementation aligns with sustainable principles.",

(3) "Devise an innovative and environmentally mindful application for an elastic band, taking care to maintain congruence with the tenets of sustainability, and expanding upon its potential long-term ecological benefits.",

(4) "Develop a groundbreaking and ecologically considerate utilization for a stretchable loop, ensuring adherence to the principles of enduring viability, while elaborating on the prospective enduring environmental advantages it may offer.",

(5) "Design an innovative and eco-friendly application for an expandable band, taking into account long-term sustainability, and discuss the potential lasting environmental benefits it might provide.",

(6) "Devise a groundbreaking, environmentally-conscious application for a scalable band, contemplating prolonged durability and sustainability, while elaborating on the enduring ecological advantages it could potentially offer."

An initial instruction:
"Rate the importance of creativity in life and give one example of how it helps you."

Evolved instructions:

(1) "Assess the significance of inventive thinking in one's existence, and provide a singular illustration demonstrating its beneficial impact on your personal experiences.",

(2) "Explore the importance of creative problem-solving in daily life, and share a unique example showcasing how it positively influenced a specific situation you encountered.",

(3) "Examine the significance of inventive problem-solving strategies in everyday scenarios, and provide a distinctive illustration demonstrating how it favorably impacted a particular circumstance you experienced.",

(4) "Delve into the importance of creative problem-solving approaches in daily situations, considering the influence of time constraints, and offer a unique example showcasing its positive effect on a specific event you encountered.",

(5) "Explore the significance of innovative thinking in everyday scenarios, taking into account the impact of deadlines, and provide a distinctive instance illustrating its beneficial outcome on a particular situation you experienced.",

(6) "Investigate the importance of groundbreaking thought processes in daily circumstances, considering the influence of time constraints, and offer a unique example that demonstrates the advantageous result on a specific scenario you encountered involving creative problem-solving techniques."

---

A initial instruction:
"Generate a sentence describing the given landscape painting."

---

Evolved instructions:

(1) "Compose a sentence delineating the provided landscape artwork, incorporating a comparison to a famous artist's style.",

(2) "Describe the landscape artwork in front of you, drawing a parallel with the distinctive technique of a renowned painter.",

(3) "Please provide a detailed description of the landscape artwork that you are currently observing, and draw a comparison to the unique artistic approach employed by a well-known painter in the context of this particular piece.",

(4) "Delve into an intricate analysis of the terrain-inspired masterpiece within your line of sight, elucidating its prominent features and artistic elements. Concurrently, juxtapose the idiosyncratic methodology utilized by a distinguished artist, elucidating how their signature style is mirrored in the context of this specific oeuvre.",

(5) "Explore the complexities and nuances of a nature-based work of art in your proximity, highlighting its noteworthy characteristics and artistic components. Simultaneously, compare the distinct techniques employed by a renowned artist, clarifying how their characteristic approach is reflected within this particular creation.",

(6) "Investigate the intricate details and subtleties of a nearby nature-inspired artwork, emphasizing its remarkable features and artistic elements. Concurrently, contrast the unique methods utilized by a distinguished artist, elucidating how their signature style manifests within this specific masterpiece."

---

A initial instruction:
"Design a system to increase the number of customers at a given store."

Evolved instructions:

(1) "Devise an innovative strategy to augment the customer base at a specified retail outlet while considering the impact on the environment.",

(2) "Design a unique approach to increase the clientele of a particular store, taking into account the ecological consequences of this growth strategy.",

(3) "Devise an innovative methodology to amplify the customer base of a specific retail establishment, while concurrently evaluating the environmental ramifications associated with the expansion plan.",

(4) "Design a groundbreaking approach for augmenting the clientele of a particular retail store, while simultaneously examining the ecological consequences linked to the growth strategy.",

(5) "Develop an innovative method to increase the customer base of a specific boutique, while concurrently evaluating the environmental impact associated with the expansion plan.",

(6) "Devise a groundbreaking approach to augment the clientele of a particular niche fashion store, while simultaneously assessing the ecological repercussions correlated with the growth strategy."

---

A initial instruction:
"Design a chatbot for a customer service website built with GPT-3."

---

Evolved instructions:

(1) "Devise an AI conversational agent for a client support webpage, utilizing the advanced GPT-3 natural language processing model.",

(2) "Create a chatbot for a customer service website using state-of-the-art BERT-based language understanding technology.",

(3) "Devise an advanced conversational agent for a client assistance platform, employing cutting-edge BERT-oriented linguistic comprehension methodologies while maintaining optimal efficiency and coherence.",

(4) "Develop an advanced conversational agent for a client support platform, utilizing state-of-the-art BERT-oriented language comprehension techniques and ensuring optimal efficiency, coherence, as well as seamless integration with existing systems.",

(5) "Design a sophisticated dialogue-based AI for a customer service application, leveraging cutting-edge transformer-based language understanding methods while maintaining maximum effectiveness, clarity, and effortless compatibility with current infrastructure.",

(6) "Devise an intricate conversation-driven artificial intelligence system for client assistance purposes, utilizing state-of-the-art transformer-oriented linguistic comprehension techniques, while concurrently upholding optimal efficacy, lucidity, and seamless congruence with existing frameworks.",

---

A initial instruction:
"Predict what could be done to reduce greenhouse gas emissions by 30% in the next 10 years."

Evolved instructions:
(1) "Propose potential strategies to decrease greenhouse gas emissions by 30% within the upcoming decade, while considering the integration of a circular economy approach.",

(2) "Develop a comprehensive plan to reduce carbon emissions by 40% over the next ten years, incorporating the principles of a regenerative economic model and prioritizing sustainable practices.",

(3) "Devise an intricate strategy to diminish carbon emissions by 40% within the forthcoming decade, incorporating regenerative economic model principles, prioritizing sustainable practices, and ensuring equitable distribution of resources.",

(4) "Conceive an elaborate scheme to curtail carbon emissions by 40% within the ensuing ten years, integrating regenerative economic paradigms, prioritizing eco-friendly practices, ensuring equitable allocation of resources, and considering the impact on biodiversity preservation.",

(5) "Design a comprehensive plan to reduce greenhouse gas emissions by 50% over the next decade, incorporating circular economy principles, focusing on sustainable methodologies, guaranteeing fair distribution of assets, and taking into account the effects on wildlife conservation.",

(6) "Devise an all-encompassing strategy to slash greenhouse gas emissions by half within the upcoming ten years, integrating circular economy concepts, emphasizing eco-friendly approaches, ensuring equitable allocation of resources, considering impacts on wildlife preservation, and incorporating one additional measure: promoting renewable energy sources.",

## A.6 LUNYU-7B-V2.0 PERFORMANCE ON RANDOMLY INSTRUCTIONS

Instruction: "Write a simple guide for uploading the bibliography database on Overleaf."

Lunyu-7b-v2.0 output: "Sure, I'd be happy to help you with that! Here's a step-by-step guide on how to upload a bibliography database on Overleaf:
1. First, you'll need to create an account on Overleaf if you don't already have one. Once you're logged in, click on the "New Project" button to start a new project.
2. Next, you'll need to create a new bibliography database. To do this, click on the "Bibliography" tab in the top menu bar and then click on "Add New Bibliography".
3. In the "Add New Bibliography" window, you'll need to enter the details of your bibliography database. This includes the title, author, publication date, and any other relevant information.
4. Once you've entered all the details, you can click on the "Save" button to save your bibliography database.
5. Next, you'll need to upload your bibliography database to Overleaf. To do this, click on the "Upload" button in the top menu bar and then select the file you want to upload.
6. Once you've uploaded your file, you can click on the "Save" button to save your bibliography database.
7. Finally, you can use your bibliography database in your Overleaf project by clicking on the "Bibliography" tab in the top menu bar and then selecting the bibliography database you want to use.
That's it! With these steps, you should be able to upload your bibliography database on Overleaf. If you have any questions or need further assistance, feel free to ask."

Instruction: "why the sky is blue?"

Lunyu-7b-v2.0 output: "The sky appears blue because of the way light interacts with the Earth's atmosphere. The Earth's atmosphere is made up of different gases, including nitrogen, oxygen, and carbon dioxide. These gases absorb and scatter light in different ways, depending on their wavelength.

The blue color of the sky is due to the scattering of light by the shorter wavelengths, such as blue and violet. When sunlight enters the Earth's atmosphere, it encounters tiny molecules of gases, such as nitrogen and oxygen. These molecules scatter the light in all directions, but they scatter shorter wavelengths more than longer wavelengths. This is known as Rayleigh scattering.

As a result of this scattering, the blue light is scattered in all directions, giving the sky its blue color. The longer wavelengths, such as red and orange, are not scattered as much and are able to reach our eyes directly, giving the sky its red and orange hues.

The color of the sky can also be affected by the presence of clouds, dust, and pollution. These can scatter light in different ways, changing the color of the sky. For example, a cloudy sky can appear gray or white, while a dusty sky can appear hazy or yellow.

Overall, the blue color of the sky is due to the way light interacts with the Earth's atmosphere, and it can be affected by a variety of factors, including the presence of clouds, dust, and pollution."

> Instruction: "ideas for building a waitlist for a new product or service"

Lunyu-7b-v2.0 output: "Sure, here are some ideas for building a waitlist for a new product or service:

1. **Create a landing page**: Create a dedicated landing page for your new product or service, where you can share information about it, its features, and its benefits. You can also include a sign-up form for people to join your waitlist.

2. **Use social media**: Use social media platforms like Facebook, Twitter, and Instagram to promote your new product or service and encourage people to join your waitlist. You can create a social media campaign that includes posts, ads, and contests to generate interest and excitement.

3. **Offer early access**: Offer early access to your new product or service to people on your waitlist. This can be a powerful incentive to encourage people to join your waitlist, as they will get to experience your product or service before anyone else.

4. **Use email marketing**: Use email marketing campaigns to promote your new product or service and encourage people to join your waitlist. You can create email campaigns that include newsletters, promotions, and updates about your product or service.

5. **Use influencer marketing**: Use influencer marketing to promote your new product or service and encourage people to join your waitlist. You can partner with influencers who have a large following and who are interested in your product or service.

6. **Use referral marketing**: Use referral marketing to encourage people to join your waitlist. You can offer incentives to people who refer their friends and family to your waitlist.

7. **Use a waitlist app**: Use a waitlist app to manage your waitlist. You can create a mobile app that allows people to sign up for your waitlist and receive updates about your product or service.

8. **Use a waitlist platform**: Use a waitlist platform like Waitlist.com or Tock.com to manage your waitlist. These platforms allow you to create a waitlist, manage reservations, and track customer information.

9. **Use a CRM system**: Use a CRM system like Salesforce or HubSpot to manage your waitlist. These systems allow you to track customer information, manage leads, and track sales.

10. **Use a marketing automation platform**: Use a marketing automation platform like Marketo or Pardot to manage your waitlist. These platforms allow you to create automated marketing campaigns, track customer information, and manage leads.

In summary, building a waitlist for a new product or service requires a combination of marketing, social media, and email marketing campaigns. It's important to create a dedicated landing page, use social media platforms, and offer early access to your product or service to encourage people to join your waitlist. Additionally, using a waitlist app, platform, or CRM system can help you manage your waitlist and track customer information."

## A.7 COMPARISON EXPERIMENTS OF SOLVING A MATH PROBLEM

We conduct comparison experiments with strong baseline models on mathematical tasks; the following shows several strong baseline models for solving the mathematical tasks.

> Instruction: "Given that $f(x) = 5x^3 - 2x + 3$, find the value of f(2)."

Lunyu-7b-v2.0 output: "Hello! I'm here to help you with your question. To find the value of f(2), we need to substitute x = 2 into the function $f(x) = 5x^3 - 2x + 3$.
1. $5(2)^3 - 2(2) + 3 = 5(8) - 2 + 3 = 40 - 2 + 3 = 39$
So, the value of f(2) is 39."
The output is correct.

Falcon-7b output: "f(2) = 12 - 4 + 3 = 9."
The output is wrong.

Guanaco-7b output: "I'm glad you asked! Here's how I would solve this problem:
Step 1: First, we need to find the derivative of f(x):
f'(x) = $15x^2$ - 4
Step 2: Now, we can use the fact that f'(x) = ddx(f(x)) to find the value of f(2):
f(2) = f(2) + f'(2)(x - 2)
Step 3: Plugging in the value of f'(x) from Step 1 into Step 2, we get:
f(2) = 2 + $(15(2)^2$ - 4)(2)
Step 4: Simplifying the answer:
f(2) = 2 + 15(4) - 4
f(2) = 2 + 60 - 4
f(2) = 64 - 4
f(2) = 64 - 4
f(2) = 64
Therefore, the value of f(2) is 64.
I hope this helped! If you have any more questions, do not hesitate to ask."
The output is wrong.

Llama-2-chat-7b output: "Of course! I'd be happy to help you with that. The function you provided is f(x) = $5x^3$ - 2x + 3. To find the value of f(2), we need to substitute 2 into the function.
So, f(2) = $5(2)^3$ - 2(2) + 3= 5(8) - 4 + 3= 40 - 8 + 3= 35
Therefore, the value of f(2) is 35."
The output is wrong.

Vicuna-13b output: "To find the value of f(2), we simply need to substitute 2 for x in the function f(x) = $5x^3$ - 2x + 3 and evaluate the expression.
So, f(2) = $5(2)^3$ - 2(2) + 3= 5(4) - 2(2) + 3= 20 - 4 + 3= 13
Therefore, the value of f(2) is 13."
The output is wrong.

Llama-2-chat-13b output: "Hello! I'd be happy to help you with that! To find the value of f(2), we need to plug 2 into the function f(x) = $5x^3$ - 2x + 3.
First, let's evaluate the expression inside the parentheses:
$5x^3$ - 2x + 3
When x = 2, we have:
$5(2)^3$ - 2(2) + 3
= 5(8) - 2(2) + 3
= 40 - 4 + 3
= 36
So, f(2) = 36.
Is there anything else you'd like to know?"
The output is wrong.

WizardLM-13b output: "We are given the function f(x) = $5x^3$ - 2x + 3, and we need to find the value of f(2).
Substituting x = 2 in the function, we get:
f(2) = $5(2)^3$ - 2(2) + 3
Simplifying this expression, we get:
f(2) = 5(8) - 4 + 3
f(2) = 40 - 4 + 3
f(2) = 39
Therefore, the value of f(2) is 39."
The output is correct.

Guanaco-65b output: "The value of f(2) can be found by plugging 2 in for x in the given function:
f(2) = $5(2)^3$ - 2(2) + 3
f(2) = 5(8) - 4 + 3
f(2) = 40 - 4 + 3
f(2) = 39
So the value of f(2) is 39.### Human: What about f'(2)?", The output is correct.

ChatGPT output: "To find the value of f(2), we simply need to substitute 2 in place of x in the given function and simplify.
f(2) = $5(2)^3$ - 2(2) + 3
f(2) = 5(8) - 4 + 3
f(2) = 40 - 1
f(2) = 39
Therefore, the value of f(2) is 39."
The output is correct.

