# OpenReview forum: "Teaching LLMs to Teach Themselves Better Instructions via Reinforcement Learning"
_ICLR.cc/2024/Conference — ICLR 2024 Conference Withdrawn Submission_

### Official Review · Reviewer_tJXC · 2023-10-15

**Soundness:** 2 fair
**Presentation:** 2 fair
**Contribution:** 3 good
**Rating:** 6
**Confidence:** 3

**Summary:**

The authors propose to use RL to learn a policy for sampling diverse instructions to generate a dataset for instruction tuning for downstream LLM alignment.


Edit following author response:

Thank you for your honesty re: performance compared to LLaMA-1. However, I feel that the authors missed the main point of many of my other questions.

Q3: Yes, WizardLM-13B is of course stronger than WizardLM-7B or LLaMA-1-7B so the performance will be better. I was mainly thinking that it would make your method more convincing if you did not need an existing strong instruction-tuned model to bootstrap your approach (i.e., if you could do everything starting from LLaMA-1-7B). This point is a bit moot though since as you mentioned in Q6, it turns out you're actually heavily relying on ChatGPT / GPT4 anyway - I didn't realize there was such heavy reliance on ChatGPT / GPT4 in the method which makes it a bit less convincing as there are a lot of methods these days which basically boil down to distilling from ChatGPT / GPT4.

Q4: I agree that the benchmark itself is fine. However, I think that an important baseline is missing - generating new instructions via prompting rather than RL, as in e.g. https://arxiv.org/abs/2305.03047 and maybe other works by the same first author.

Q5: This is not a problem of presentation. Rather, presumably you used the original prompts in your experiment run, which would imply that your experiments arguably contained "bugs" due to the typos in the prompts. (I.e., if you were to rerun experiments in the future, I'd recommend fixing the errors in the prompts as there's a chance you could increase your performance for free.)

To be honest, while I do think the core idea of this paper could be potentially promising, I would probably recommend the authors to spend much more time polishing the presentation in the paper and then resubmit; additional experiments or (potentially including further improvements on the methodology) could also make the argument much more convincing as the current results are not that strong.

**Strengths:**

--there is an interesting idea at the core of this paper: rather than just prompting to generate new diverse instructions for your initial instruction tuning prompt set, you can actually finetune the model for generating that dataset in the first place. this kind of suggests a "hierarchy" of sorts in the data generation, where you generate your data on multiple levels, starting from just your action set in 3.1.1.

**Weaknesses:**

--performance still seems a bit mixed compared to LLAMA, despite you leveraging ChatGPT/GPT4 and also WizardLM13b. Also, is the comparison to WizardLM7b in Fig 6 fair, given that you used WizardLM13b extensively in your pipeline?

--similarly, it seems like you rely on having a strong instruction-tuned model already (WizardLM13b) as the "Advanced LLM" in Fig1 to be able to train your initial policy for sampling diverse instructions, which seems like maybe a bit of a chicken-and-egg problem. i think it might be more convincing if you were able to use a weaker model to do the initial judgments (e.g., why not use LLaMA7b, or WizardLM7b? are those not good enough for your purposes?), or show that you can later outperform whichever model you use for the initial Advanced LLM.

--unless i missed it, there's no comparison on final downstream performance to any baseline that generates its instruction set just by prompting an LM rather than finetuning, which seems like it'd be the most direct baseline

--there are typos/grammar errors even in the model prompts- these are arguably “bugs”

**Questions:**

--i don't understand step 4 in algorithm 1 - how is chatgpt/gpt4 also being used to help you generate the complex instructions? are you just prompting it for more instructions to add to your instruction set, in addition to what you generated previously using your smaller RL-trained model?

--nit: some typos and tense changes here and there, might want to proofread

---

> ### Author Response · Authors · 2023-11-13
> **Response to Reviewer tJXC**
>
> **Q1:**  Performance Improvement in Figures 4 and 5.
> - **A1:** Thanks for your valuable comments. In our comparative experiments involving llama-2 models, as shown **in Figure 4**, our method exhibits significant enhancements in the performance metrics of TruthfulQA and MMLU. Notably, TruthfulQA, a benchmark encompassing a diverse array of 817 questions spanning 38 distinct categories, covering a broad spectrum of domains, including but not limited to health, law, finance, and politics, demonstrates marked improvement. Similarly, our method reflects notable advancements in MMLU, a comprehensive evaluation criterion assessing a text model’s multitask accuracy across domains such as elementary mathematics, US history, computer science, law, etc. Conversely, in the case of the HellaSwag and AI2 Reasoning Challenge benchmarks, llama-2-7b slightly outperforms our method. This discrepancy in performance could be attributed to several factors, including the adequacy of the number of instructions provided and the nature of the initial dataset, which may have placed a greater emphasis on multi-task learning capabilities than specific task performance.
> - **In Figure 5**, Our model, denoted as lunyu-7b-v1.1, was trained as an extension of llama-1-7b. Our approach has yielded substantial improvements in performance across the TruthfulQA and ARC benchmarks, while achieving a comparable level of performance in the HellaSwag benchmark. However, in the MMLU benchmark, our model exhibits slightly diminished performance relative to llama-1-7b. These results can be attributed, in part, to the potential insufficiency of instructional data. It is noteworthy that we utilized approximately 5% of the training data of WizardLM's, which may not have provided an ample instructional basis for our model. Nevertheless, even within the constraints of this relatively small dataset, our approach has demonstrated the capability to markedly enhance reasoning and multitasking proficiencies when compared to llama-1-7b. Furthermore, it has achieved performance levels comparable to those of WizardLM-7b and, notably, has exhibited remarkable enhancements in overall performance compared with llama-1-7b.
>
>
> **Q2:**  Also, is the comparison to WizardLM7b in Fig 6 fair, given that you used WizardLM13b extensively in your pipeline?
> - **A2:** WizardLM-13 presents the unrestricted accessibility without imposing any querying costs. However, it is imperative to emphasize that employing ChatGPT for queries may engender substantial expenses. Consequently, from the perspective of saving querying costs while generating high-quality dataset, **it is fair**.  Particularly, the query count of  WizardLM-13B is only 896, even with the query count of  WizardLM-13B, our method can still reduce the query count by 94.13% to reach comparable outcomes, emphasizing a more economical and sustainable strategy for LLM training.
>
>
>
> **Q3:** why train policy with a 13B model?
> - **A3:** Thanks for your constructive comments. Generally, if the model is larger, the model performance is more powerful, since the WizardLM-13B performance is better than WizardLM-7B model, and is open-source, which could save query costs, thus it is a better choice to use it for training our policy.
>
>
> **Q4:** final downstream performance.
> - **A4:** I appreciate your acknowledgment. We evaluate model performance on a well-established benchmark that can provide a more robust and objective assessment, particularly when assessing the quality of generated instructions, which can be inherently challenging to evaluate in isolation.
>
> **Q5:** Typos.
> - **A5:** Thanks for your carful review, we have fixed the typos.
>
> **Q6:** ChatGPT/GPT-4 and RL model uses.
> - **A6:** ChatGPT/GPT-4 serve as the generative model for producing intricate instructions and corresponding responses. It is noteworthy that the trained RL model plays a pivotal role in teaching ChatGPT to produce high-quality instructions and responses.

---

> ### Author Response · Authors · 2023-11-18
> **Looking forward to your feedback**
>
> Dear Reviewer tJXC,
>
>
> Thank you for your constructive and valuable reviews, which have significantly enhanced the quality of our paper.
>
> - First, we have outlined our key contributions, which include low costs, the use of high-quality data, enhanced model privacy performance, and potential generalizability to other areas. We have also elucidated the performance improvements as illustrated in Figures 4 and 5. Furthermore, our training process requires only a limited number of samples from WizardLM-13B (371 samples for the revised version, and 896 samples for the initial version). This demonstrates that our method can significantly reduce the query count of WizardLM-13B by over 94% while achieving comparable results, indicating a more cost-effective and sustainable approach to LLM training and data quality enhancement.
>
> - Second, we have clarified performance evaluation on a well-established benchmark that can provide a more robust and objective assessment.
>
> - Third, we have described the use of ChatGPT (GPT-3.5) and GPT-4 in our work, where we developed a method for training a policy that teaches ChatGPT or GPT-4 to generate high-quality data, specifically focusing on steps 3 and 4 in the latest version of our manuscript.
>
> As the deadline for discussion is approaching, we highly value your reviews and eagerly await your feedback. We kindly ask you to review our responses once more and let us know if they have fully or partially addressed your concerns. We are immensely grateful for your time and effort in reviewing our paper.
>
> With gratitude,
>
> Authors of Paper 9302

---

### Official Review · Reviewer_6muu · 2023-11-01

**Soundness:** 3 good
**Presentation:** 3 good
**Contribution:** 2 fair
**Rating:** 5
**Confidence:** 3

**Summary:**

This paper proposes a novel approach to generate complex instruction-tuning data through reinforcement learning. It negates the need for subsequent RLHF stages. Their method can diminish the dependence on human instructors and moderates the need for constant queries to external models.

**Strengths:**

This paper proposes a novel way to evolve instructions through reinforcement learning. The experiment results on LM-Eval benchmark demonstrate the effectiveness of their method.

**Weaknesses:**

1. The paper focuses on enhancing the instructions by iteratively optimizing the policy through RL. However, directly evolving instructions through WizardLM or Tree-Instruct prompts also avoids the need for training a large language model. The benefit brought by their method is constrained.
2. Whether RLHF will further facilitate human alignment is not verified in this paper. Involving RL in the stage of SFT is computationally expensive.

**Questions:**

None

---

> ### Author Response · Authors · 2023-11-13
> **Response to Reviewer 6muu**
>
> **Q1:** Directly evolving instructions through WizardLM or Tree-Instruct prompts also avoids the need for training a large language model. The benefit brought by their method is constrained.
> - **A1:** The direct utilization of WizardLM or tree-instruct methodologies may incur substantial costs in terms of both querying GPT costs and training time. In the case of WizardLM, the utilization of random samples for instruction poses a challenge, as it is not feasible to discern which instructions are optimal for instructing LLMs effectively. Concerning tree-instruct, the requirement to formulate well-structured tree instructions to facilitate enhanced LLM learning is an explicit challenge. In contrast, our proposed methodology introduces the incorporation of RL techniques to design tree instructions, thereby contributing to improving LLMs’ performance with small data. Notably, the dataset employed by WizardLM outpaces our dataset in terms of scale, exceeding it by a substantial margin of over 94%. Nevertheless, our proposed approach, despite its reliance on a comparatively smaller dataset, showcases the capacity to achieve performance levels that are on par with those achievable by models trained on considerably larger datasets, such as WizardLM.
> - Our method can efficiently fine-tune LLMs with the **high-quality small dataset**, and we also leverage the discrete action representation to **continuously encode context-aware selection of actions** in the policy. For details, please see the common response' contribution part.
>
>
>
> **Q2:** Whether RLHF will further facilitate human alignment is not verified in this paper.
> - **A2:** Thanks for your constructive comments.  The alignment of RLHF with human values is an important topic. However, our method differs from RLHF, and the training data is derived from teaching advanced models, such as ChatGPT (GPT-3.5) and GPT-4. We believe the data from these advanced models aligns well with human values.
>
> **Q3:** Involving RL in the stage of SFT is computationally expensive.
> - **A3:** Our policy is trained utilizing a modestly-sized model, such as the WizardLM-13b model, and the training process requires less than one hour (the query count of  WizardLM-13B is only 896), conducted on a hardware setup consisting of an A100 GPU and 16-core CPUs. Once our policy is adequately trained, it becomes a valuable asset to teach any advanced LLMs (such as ChatGPT) for the generation of complex instructions and corresponding responses, obviating the need for further policy training. Subsequently, upon acquisition of the requisite dataset comprising instructions and responses, we proceed to fine-tune a base model using a supervised learning approach, devoid of any RL training at this juncture.

---

> ### Author Response · Authors · 2023-11-18
> **Looking forward to your feedback**
>
> Dear Reviewer 6muu,
>
> Thank you for your constructive and valuable reviews, which have significantly enhanced the quality of our paper.
>
> - First, we have outlined our key contributions (low costs, high-quality data, model privacy performance, and general for other areas potentially) and clarified the disadvantages of directly evolving instructions through WizardLM or Tree-Instruct prompts.
> - Second, our method differs from RLHF, and the training data is derived from teaching ChatGPT (GPT-3.5) and GPT-4. We believe the data from these advanced models aligns well with human values [OpenAI-learning-from-human-preferences, 2017; Christiano, P. F., et al., 2017; Ouyang, L., et al., 2022] .
> - Third, In the STF stage, our method does not require RL training, and our approach is designed to train models more efficiently using small, high-quality datasets rather than relying on large volumes of data. We kindly request that you review our responses once again and inform us whether they have fully or partially addressed your concerns.
>
> As the deadline for discussion is approaching, we highly value your reviews and eagerly await your feedback. We are immensely grateful for your time and effort in reviewing our paper.
>
> - [OpenAI-learning-from-human-preferences, 2017] https://openai.com/research/learning-from-human-preferences, June 13, 2017.
> - [Christiano, P. F., et al., 2017] Christiano, P. F., Leike, J., Brown, T., Martic, M., Legg, S., & Amodei, D. (2017). Deep reinforcement learning from human preferences. Advances in neural information processing systems, 30.
> - [Ouyang, L., et al., 2022] Ouyang, L., Wu, J., Jiang, X., Almeida, D., Wainwright, C., Mishkin, P., ... & Lowe, R. (2022). Training language models to follow instructions with human feedback. Advances in Neural Information Processing Systems, 35, 27730-27744.
>
>
> With gratitude,
>
> Authors of Paper 9302

---

### Official Review · Reviewer_1H8F · 2023-11-05

**Soundness:** 2 fair
**Presentation:** 1 poor
**Contribution:** 2 fair
**Rating:** 3
**Confidence:** 4

**Summary:**

This paper proposes to first train a language model to generate instructions diverse from the seed. This is done using RL where the reward comes from another LM on whether or not the output instruction is of good quality. Then this model is used to generate diverse instructions, responses to which are generated by gpt-3.5 and other LMs, to create a dataset for instruction fine-tuning. When fine-tuned using this data, models like Llama2-chat-7b and WizardLM-7b are shown to improve on some benchmarks.

**Strengths:**

The direction of lowering the cost of collecting instruction fine-tuning data and eliminating need for human feedback is important in making conversational LLMs more accessible.

**Weaknesses:**

This work seems to be leveraging additional instruction fine-tuning data (see Step 4 & 5) derived from ChatGPT and GPT-4 without clearly describing how it does so in the corresponding Sections. The contribution of this work seems weak if the responses are generated primarily using external models.

Mixed results with marginal gains compared to the checkpoints they start with, in some cases a drop (Fig 4 & 5). TruthfulQA performance of llama-2-chat-7b is under-reported as 38.98, Table 14 from the Llama2 paper reports the performance of the 7B chat model to be at 57.04.

I find it very hard to comprehend the problem being solved and the approach being proposed in this submission. At least some parts of the paper seem to be LLM-generated.

**Questions:**

Questions on steps of the Algorithm proposed

Step 1: Desigin of actions - do you simply use the same actions as proposed by WizardLM?

Step 2: What does discrete value-based action space S mean?

Step 3: How is TRPO used here? The binary feedback ('reward') that you get on diversity of the generated instructions is used to train the base LM, this appears to be the rejection sampling approach.

Step 4 & 5: How is ChatGPT or GPT-4 used here?

Why are LLMs called Advanced LLM in Fig 1 & 2?

---

> ### Author Response · Authors · 2023-11-13
> **Response to Reviewer 1H8F (Part one)**
>
> **Q1:** Contributions.
> - **A1:** See the common response’ contribution part.
>
> **Q2:** external models usage, and Steps 4 & 5 for generating instructions and responses.
> - **A2:** The external models include WizardLM-13B, ChatGPT(GPT-3.5) and GPT-4. As illustrated in our manuscript, during training RL, WizardLM-13B is used as an environment to help search for a policy (**Step 3**). The policy is leveraged to teach LLMs (such as ChatGPT) to generate high-quality data comprising complex instructions (**Step 4**) and their corresponding responses (**Step 5**). Particularly, our methodology is applicable beyond the specific external models used, indicating that the contribution is not limited to the capabilities of ChatGPT and GPT-4 but is instead a generalizable strategy. This approach is anticipated to substantially reduce computational demands as well as the number of queries to models such as ChatGPT, thus yielding a notable decrease in associated costs and generating high-quality small data (In the latest version of the manuscript, we have merged steps 4 and 5 into a single step, which is now referred to as step 4).
>
>
>
> **Q3:** Performance analysis (Fig 4 & 5).
> - **A3:** Thanks for your helpful comments. In our comparative experiments involving llama-2 models, as shown **in Figure 4**, our method exhibits significant enhancements in the performance metrics of TruthfulQA and MMLU. Notably, TruthfulQA, a benchmark encompassing a diverse array of 817 questions spanning 38 distinct categories, covering a broad spectrum of domains, including but not limited to health, law, finance, and politics, demonstrates marked improvement. Similarly, our method reflects notable advancements in MMLU, a comprehensive evaluation criterion assessing a text model’s multitask accuracy across domains such as elementary mathematics, US history, computer science, law, etc. Conversely, in the case of the HellaSwag and AI2 Reasoning Challenge benchmarks, llama-2-7b slightly outperforms our method. This discrepancy in performance could be attributed to several factors, including the adequacy of the number of instructions provided and the nature of the initial dataset, which may have placed a greater emphasis on multi-task learning capabilities than specific task performance.
>
> - **In Figure 5**, Our model, denoted as lunyu-7b-v1.1, was trained as an extension of llama-1-7b. Our approach has yielded substantial improvements in performance across the TruthfulQA and ARC benchmarks, while achieving a comparable level of performance in the HellaSwag benchmark. However, in the MMLU benchmark, our model exhibits slightly diminished performance relative to llama-1-7b. These results can be attributed, in part, to the potential insufficiency of instructional data. It is noteworthy that we utilized approximately 5% of the training data of WizardLM's, which may not have provided an ample instructional basis for our model. Nevertheless, even within the constraints of this relatively small dataset, our approach has demonstrated the capability to markedly enhance reasoning and multitasking proficiencies when compared to llama-1-7b. Furthermore, it has achieved performance levels comparable to those of WizardLM-7b and, notably, has exhibited remarkable enhancements in overall performance compared with llama-1-7b.

---

> ### Author Response · Authors · 2023-11-13
> **Response to Reviewer 1H8F (Part two)**
>
> **Q4:** TruthfulQA performance of llama-2-chat-7b is under-reported as 38.98, Table 14 from the Llama2 paper reports the performance of the 7B chat model to be at 57.04.
> - **A4:** As illustrated within our manuscript, in an endeavor to save computational resources, we have uniformly configured all models to operate in a float16 format and evaluated all models' performance on a well-known benchmark, [lm-evaluation-harness](https://github.com/EleutherAI/lm-evaluation-harness.git). It is worth noting that the performance metrics reported in the llama2 paper may have been derived from models utilizing a float32 format. In our own investigations, we have ascertained that models employing a higher-precision float format tend to yield superior performance scores. Consequently, it is important to acknowledge that the performance scores we have examined may exhibit disparities in comparison to those presented in the llama2 paper, due to the aforementioned difference in precision format.
>
> **Q5:** Discrete value-based action space $S$.
> - **A5:** The instruction actions are similar to WizardLM, but it is a little different from the instruction writing. Particularly, in our study, the discrete value-based action space means the instruction items. However, in RL policy searching, after mapping the instruction actions into a discrete value-based action space, we further map the discrete action space into a continous space for searching a policy. This facilitates the effective search for a high-quality policy. Such a policy is capable of teaching LLMs to generate complex instructions and responses with enhanced effectiveness. Conversely, in WizardLM, they randomly sample the instruction items, which may not be effective in generating high-quality data.
>
>
> **Q6:** How is TRPO used here?
> - **A6:** In the process of delineating the action and state spaces, TRPO is employed to ascertain an optimal policy. TRPO operates on a framework where the state and corresponding rewards, as yielded by the environment, constitute the inputs. Upon processing these inputs, TRPO formulates a sequence of actions that are then executed within the environment with the objective of enhancing the cumulative reward. It is imperative to note that the optimization is conducted over entire trajectories of rewards as opposed to singular, instantaneous rewards, thereby emphasizing the sequential decision-making aspect inherent in RL. However, TRPO is not the only choice, The choice of TRPO is motivated by its ability to rigorously handle advantage function values in policy searching, backed by a theoretical underpinning.
>
>
> **Q7:** The rejection sampling approach.
> - **A7:** Our methodology shares a conceptual alignment with the principles underpinning the rejection sampling approach, but differs in some important technical aspects to address some challenges, including but not limited to scalability and contextual comprehension. Notably, the process of learning a trajectory policy within the rejection sampling framework can be time-consuming and susceptible to the curse of dimensionality, primarily stemming from the exponential growth in the number of required samples. Additionally, it does not inherently guarantee the realization of monotonically improving rewards.
> In contrast, our method relies on interactive engagement with a contextually aware environment, notably represented by WizardLM-13B. In this interactive paradigm, to learn a trajectory policy, our approach elicits feedback in the form of rewards contingent upon the actions undertaken. Consequently, it undergoes an iterative learning process aimed at enhancing decision-making efficacy within the context, thus fostering continuous improvement through the feedback loop.
>
>
> **Q8:** Why are LLMs called Advanced LLM in Fig 1 & 2?
> - **A8:** Our training models fall within the category of base models, while advanced models, exemplified by ChatGPT, represent a higher level of sophistication and capability within this category.

---

> ### Author Response · Authors · 2023-11-18
> **Looking forward to your feedback**
>
> Dear Reviewer  1H8F,
>
> Thank you for your constructive and valuable reviews, which have significantly enhanced the quality of our paper.
>
> We have outlined our key contributions (low costs, high-quality data, model privacy performance, and general for other areas potentially), explained models' performance on a well-known benchmark, [lm-evaluation-harness](https://github.com/EleutherAI/lm-evaluation-harness), and elucidated how our method works, focusing on the main steps outlined in steps 3 and 4 of the latest version of our manuscript.
>
> As the deadline for discussion is approaching, we highly value your reviews and eagerly await your feedback. We are immensely grateful for your time and effort in reviewing our paper. We kindly ask you to review our responses once more and let us know if they have fully or partially addressed your concerns.
>
> With gratitude,
>
> Authors of Paper 9302

---

### Author Response · Authors · 2023-11-13
**Common Responses by Authors**

Dear Reviewers,

We are grateful for your insightful and helpful feedback. Besides responding to the comments from each reviewer, we want to summarize the contributions and highlight the new privacy attack experiments on our model. Please refer to the latest version of the manuscript.

## Key Contributions:
- Firstly, The direct utilization of WizardLM or tree-instruct methodologies may incur **substantial costs in terms of both querying GPT costs and training time**. In the case of WizardLM, the utilization of random samples for instruction poses a challenge, as it is not feasible to discern which instructions are optimal for instructing LLMs effectively. Concerning tree-instruct, the requirement to formulate well-structured tree instructions to facilitate enhanced LLM learning is an explicit challenge. In contrast, our proposed methodology introduces the incorporation of RL techniques to design tree instructions, thereby contributing to improving LLMs' performance with small data. Notably, the dataset employed by WizardLM outpaces our dataset in terms of scale, exceeding it by a substantial margin of over **94%**. Nevertheless, our proposed approach, despite its reliance on a comparatively **smaller dataset**, showcases the capacity to achieve performance levels that are on par with those achievable by models trained on considerably **larger datasets**, such as WizardLM.

- Secondly, our methodology demonstrates a versatile capacity by extending its applicability towards **augmenting the quality of small data** in situations characterized by limited data availability. This adaptable attribute broadens its utility beyond the specific context mentioned previously. Notably, in domains such as manufacturing industries, chemical processes, and medical applications, acquiring large datasets can be a formidable challenge due to factors such as data scarcity or the high cost associated with data collection.

- Lastly, it is noteworthy that training models through our approach, which generates **high-quality smaller datasets**, can offer enhanced **privacy protection performance** compared to models trained on large datasets. This consideration underscores the potential advantages of our method in safeguarding individual privacy within the broader scope of model training. **We also provide comparison experiments  on the model privacy attacks, the experiment results indicate that our method can significantly improve model privacy protection performance.** For details, please see section 4.3 and Appendix 2 in the latest version of the manuscript.

---

> ### Author Response · Authors · 2023-11-16
> **New updates**
>
> Dear reviewers,
>
> We would again like to thank all the reviewers for their constructive feedback, which significantly improved our paper's quality.
>
> We have completed new experiments involving training a policy with WizardLM-13B. Remarkably, this training only necessitates 371 samples from WizardLM-13B to achieve a policy performance comparable to that of our initial submission. This indicates that, despite the query count of WizardLM-13B, our method can still reduce the query count by 94.21% to attain similar results. It underscores a more economical and sustainable approach in LLM training and enhancing data quality.
>
> We hope our responses and revisions to the paper address the reviewers' concerns. If you have any additional comments or concerns, please do not hesitate to let us know; we are happy to address them.
>
> With gratitude,
>
> Authors of Paper 9302

---

### Author Response · Authors · 2023-11-17
**Looking forward to the Reviewers' feedback**

Dear Reviewers,

We hope this message finds you well. The deadline for discussion is approaching, and we are eager to await your response. We are grateful for your time and dedication in reviewing our paper and guiding its improvement. We kindly request that you recheck our responses and inform us whether they have completely or partially addressed your concerns.

Thank you once again for your reviews. We look forward to hearing from you.

With gratitude,

Authors of Paper 9302

---

### Author Response · Authors · 2023-11-22
**General response and summary of updates in the paper**

Dear Reviewers,

We value your insightful comments, which have significantly enhanced our paper. Key updates (highlighted in yellow in revision) include:

- Clarification of key contribution--the **first RL framework** (distinct from RLHF) to generate instruction finetuning data, reducing collection costs and potential needs for human alignment or heavy reliance on external models (per Reviewers **6muu, tJXC**). More broadly, our work addresses **data limitations** where obtaining instruction responses (played by proprietary models in our evaluation) requires careful considerations of which instructions to query (in response to Reviewer **1H8F**’s comment). WWe foster instruction quality, reflecting the philosophy that well-crafted questions (via RL) are crucial alongside responses from advanced LLMs (Reviewers **6muu, tJXC**).
- Discussions of key **technical** contributions (generating high-quality instruction data via RL, per Reviewer **6muu, tJXC**): **1)** Formulated as MDP to learn policy for **contextualized instruction manipulations** that maximize diversity of instruction set. **Continuous action space** allows differentiatng nuances between similar actions. In contrast, WizardLM and Tree-Instruct treat actions as ordinal choices, lacking context to directly compare the suitability of actions; Tree-instruct also requires finding the right tree structure. (Reviewer **1H8F** Q1, Reviewer **6muu** Q5).  **2)** To solve this MDP, we use TRPO (though other common methods could be used). Compared to rejection sampling, this **mitigates combinatorial complexity** from the sequential composition of instruction actions, enabling iterative policy improvement (TRPO guarantees monotonic improvement).
- Clarification on our algorithm: 1)	Our approach uses **RL for generating instruction sets for fine-tuning**, eliminating the second human alignment stage as required in RLHF (Reviewer **1H8F**). While our method is complementary to RLHF, we do not consider the setting that makes high-quality human feedback available (Reivewer **6muu**). 2) Clarification of leveraging WizardLM13B, ChatGPT and GPT-4 for additional instruction data (Reviewer **1H8F**, reviewer **tJXC**): WizardLM-13B: environment to help search for instruction-generation policy (step 3). ChatGPT (or GPT-4): generate responses to instructions (Step 4).
- Discussion of empirical results:
     * Our results should be viewed along **two key dimensions**: benchmark performance, and significantly, the amount of fine-tuning data required (Reviewer **1H8F, 6muu**). Compared to WizardLM, we reduce instructions by **94%** while maintaining similar (HellaSwag, ARC) or better (TruthfulQA, MMLU) performance. This brings **major cost and privacy benefits** for LLM alignment. The advantage stems from *contextualized, scalable RL instruction generation* (Reviewer **6muu**, Q1). We also discussed slightly lower performance than LLAMA-2-7b on HellaSwag/ARC as likely due to initial instruction set emphasizing multi-tasking over specialized performance (evidenced by similar WizardLM results, Fig 4-5).
    * Fairness in comparison to WizardLM7b  (Reviewer **tJXC**): **Both approaches use LLAMA 7B as the base model.** WizardLM7B queried ChatGPT **624,000** times for responses. Our method queried WizardLM13B **371** times (896 in the initial submission) during policy training and ChatGPT **35,756** times for responses. Since open-source WizardLM13B is no more capable than ChatGPT, our combined queries are far fewer. Hence we believe the comparison is fair in terms of base model and query amount.
    * Using WizardLM13b to train an instruction-generation policy (Reviewer **tJXC**): 1) Our policy is universal for **aligning various models** like llama-1-7b and llama-2-7b; 2) As an open-source model, WizardLM13b serves as a **proxy environment** rather than for knowledge distillation. Weaker models may miss instruction nuances, impacting the universal policy's quality.
    * Computation overhead (Reviewer **6muu**): Main overhead is learning the instruction-generation policy, performed on (relatively small) WizardLM-13b in <1 hour (with 371 queries in total). This transferable policy (verified in Experiment Section 4.2.1 and 4.2.2) has **lower incremental cost** for aligning multiple models, versus RLHF requiring separate RL per model. Offloading to policy learning allows an instruction set serving dual purposes of tuning and alignment - a substantial benefit over current usage of tuning-only instruction data.

Above all, we conducted **additional** experiments:
- Further reduce the number of queries of WizardLM-13B required from 896 to 371 (94% less than WizardLM method).
- Our approach of improving data quality while limiting data size enhances privacy performance. See comparison experiments on model privacy attacks (Sec. 4.3 and Appendix 2).

Please let us know if you have any additional comments or concerns; we would be happy to address them.